# ORP5 and ORP8: Sterol Sensors and Phospholipid Transfer Proteins at Membrane Contact Sites?

**DOI:** 10.3390/biom10060928

**Published:** 2020-06-18

**Authors:** Nina Criado Santos, Vladimir Girik, Paula Nunes-Hasler

**Affiliations:** Department of Pathology and Immunology, Faculty of Medicine, University of Geneva, Rue Michel-Servet 1, 1211 Geneva, Switzerland; Nina.Criado@unige.ch (N.C.S.); Vladimir.Girik@unige.ch (V.G.)

**Keywords:** oxysterol binding protein like 5 and 8, OSBPL5, OSBPL8, PtdIns, PtdIns4P, PtdIns(4,5)P2, cortical endoplasmic reticulum

## Abstract

Oxysterol binding related proteins 5 and 8 (ORP5 and ORP8) are two close homologs of the larger oxysterol binding protein (OSBP) family of sterol sensors and lipid transfer proteins (LTP). Early studies indicated these transmembrane proteins, anchored to the endoplasmic reticulum (ER), bound and sensed cholesterol and oxysterols. They were identified as important for diverse cellular functions including sterol homeostasis, vesicular trafficking, proliferation and migration. In addition, they were implicated in lipid-related diseases such as atherosclerosis and diabetes, but also cancer, although their mechanisms of action remained poorly understood. Then, alongside the increasing recognition that membrane contact sites (MCS) serve as hubs for non-vesicular lipid transfer, added to their structural similarity to other LTPs, came discoveries showing that ORP5 and 8 were in fact phospholipid transfer proteins that rather sense and exchange phosphatidylserine (PS) for phosphoinositides, including phosphatidylinositol-4-phosphate (PI(4)P) and potentially phosphatidylinositol-(4,5)-bisphosphate (PI(4,5)P2). Evidence now points to their action at MCS between the ER and various organelles including the plasma membrane, lysosomes, mitochondria, and lipid droplets. Dissecting exactly how this unexpected phospholipid transfer function connects with sterol regulation in health or disease remains a challenge for future studies.

## 1. Introduction

Lipid bilayers are integral structural components of cells, allowing the compartmentalization of biomolecules and reactions that make life possible. Different intracellular compartments have distinct lipid compositions that define their identity [1]. The correct intracellular trafficking and targeting of proteins, nucleic acids, ions and metabolites, as well as replication of viruses all depend on the delicate balance and maintenance of this composition, and thus necessarily on lipid transport [1,2,3]. In addition, lipids play vital roles as signaling molecules, and rapid, dynamic, and sometimes highly localized changes in lipid composition can trigger downstream effects that allow cells to respond and adapt to daily challenges. While vesicular fission and fusion permit bulk flux and exchange of lipid between different compartments, lipid transfer proteins (LTPs) that sequester lipids in a hydrophobic pocket in order to allow their transit through the cytosol in a non-vesicular mechanism, have been increasingly recognized over the last two decades as playing an integral role in lipid metabolism and signaling [2]. 

The canonical oxysterol binding protein (OSBP) first identified in the early 80’s [4,5,6] is the founding member of the OSBP and OSBP-like (OSBPL) (also called OSBP-related, ORP) family of LTPs. As the name implies, OSBP was initially identified due to its high-affinity binding to the oxysterol 25-hydroxycholesterol and was found to be conserved across diverse species including human, insects, plants and yeast [6,7]. Based on homology mapping of the lipid binding domain of OSBP, called the OSBP homology domain (OHD) or OSBP-related domain (ORD), a multi-gene family of 12 members including OSBP and ORP1-11 was identified in mammals [7,8,9], and 7 members in the yeast *Saccharomyces cerevisiae*, termed Osh proteins [10,11]. Initial studies on Osh/ORPs focused on their role as oxysterol sensors, and they were thought to exert their influence on cellular functions ultimately through transcriptional control, albeit likely in an indirect manner that was poorly understood [10,12,13]. However, it was also clear even from early studies that Osh/ORPs could not only bind lipids other than sterols, notably phosphoinositides (PIPs), but were also involved in regulating their subcellular localization, hinting at a transport role beyond sensing. [10,11,13]. 

Notably, several Osh/ORP proteins were found early on to localize to structures now termed membrane contact sites (MCS) which are sites of close (<20 nm) apposition between membranes of two different organelles that are held together by protein tethers [13,14,15,16]. These sites provide a short cytosolic transit space ideal for non-vesicular lipid transfer. They are now recognized to serve as platforms for localized signaling, as well as accumulation and exchange of a multitude of biomolecules in addition to lipids, rendering them focal points for communication between organelles [14,15]. The solution of the structure of Osh4 revealed a resemblance to other LTPs, with a hydrophobic pocket and a lid [17]. Shortly after, its role in sterol transport was demonstrated [18], followed by a pioneering study revealing that phosphatidylinositol-4-phosphate (PI(4)P) was also transported in exchange for sterol, where it was suggested that this transport could be powered by the dephosphorylation of PI(4)P by the endoplasmic reticulum (ER)-resident phosphatase Sac1 [19]. OSBP was then was shown to transfer sterols across ER-Golgi MCS, and the dephosphorylation by Sac1 was confirmed to contribute [20]. A 4-step cycle of membrane tethering, forward sterol transfer, backward PI(4)P transfer, and PI(4)P hydrolysis was proposed to drive sterol transport up a concentration gradient, a model that was later confirmed for Osh4 [20,21]. This counter-exchange mechanism was demonstrated for other ORPs, including oxysterol binding related proteins 5 and 8 (ORP5 and ORP8) and Osh6 as discussed in further detail below. 

The proteins ORP5 and ORP8 form a subgroup within the family as their ORD sequences are most similar to each other (Figure 1). They are also the only ORPs to include a C-terminal transmembrane domain that anchors them to the membrane of the ER [7,22]. The ORP8 protein has an N-terminal acidic track that is followed by a polybasic segment and then a pleckstrin homology (PH) domain, whereas ORP5 is similar but lacks the acidic track [7,8,9,23,24,25]. These N-terminal motifs comprise distinct sites for membrane interactions. The PH domains of both ORP5/8 are atypical as they contain a 20 amino acid insertion in variable loop 3 as compared with prototypical PH domains of proteins such as AKR mouse thymoma (AKT) serine/threonine kinase and phospholipase C-gamma (PLCγ), but also of more similar proteins such a OSBP, ORP1 and ceramide transferase (CERT) as shown by sequence alignment (Figure 1F) [26]. In particular, the ORP8 PH domain lacks critical Lys and Arg residues, rendering its binding to PIPs rather weak [25,26]. The ORD, defined by its signature motif EQVSHHPP, has the form of a barrel with a hydrophobic pocket and a lid that protects an extracted lipid during transfer within an aqueous environment, typical of LTPs [17,27]. In addition, ORDs have also been shown to be able to bind two membranes simultaneously thus mediating organelle contact formation [25,26]. Interestingly, a sequence identified as an amphipathic helix (AH) within the ORD domain of ORP5 was targeted to lipid droplets (LDs) when expressed alone [26]. However, the sequence is embedded deeply inside the ORD and thus it is unclear whether it can mediate ORP5 binding to LD membranes *in vivo.* Thus, ORP5 and 8 can sense, target to membranes, and interact with lipids at multiple sites within their structure. It should be noted that there is an important distinction between lipid binding and lipid transport. Here we refer to lipid binding as any interaction between proteins and lipids that is strong or stable enough to mediate the recruitment of the proteins to a membrane. The site of this binding interaction may be on the outside surface of the protein, or within the binding pockets of lipid interaction domains such as the PH domain or the ORD. Lipid transport requires lipid binding as a prerequisite, but involves furthermore the extraction of the lipid from the membrane and transit of the bound lipid through the cytosol. The binding of PH domains to membranes generally occurs within the membrane and do not lead to extraction of the lipid, whereas lipid transport is exclusively the function of the ORD in this context.Both ORP5 and 8 are widely if not ubiquitously expressed [7,8,22,28], where ORP5 has high expression in thymus [8] and ORP8 in brain, liver, spleen, kidney, testis and placenta [7,28]. The protein ORP5 has 2 described isoforms: the longest is the canonical sequence, also called ORP5A, and the second is ORP5B, which lacks residues 134-201, disrupting part of the PH domain and thus reducing membrane association and rendering this isoform more reticular [26]. Similarly, ORP8 has 2 described splice variants, ORP8L, the canonical sequence, and a shorter ORP8S which lacks residues 1-42, rendering its N-terminus much more basic, promoting membrane association [23,24,25]. For both ORP5 and 8 tissue distribution of alternative splice forms have not been investigated, although in many studies two bands (or even three in the case of ORP8) are discerned in western blots [22,23,28,29,30,31,32], indicating that both splice forms act in concert in many tissues. 

Similar to OSBP and other ORPs, the function of ORP5 and ORP8 was initially assumed to be related to sterol sensing, but as the LTP function of other ORPs began to be described, the view on how ORP5 and 8 operate also shifted towards this direction. In this review we summarize early studies on ORP5 and ORP8 that focused on their role as oxysterol sensors, influencing important functions such as body lipid homeostasis, insulin signaling, proliferation and migration. We then describe more recent discoveries that lead to the current understanding of how ORP5 and 8 exert their function as LTPs: first influencing vesicular trafficking as putative sterol transporters, and then as phospholipid exchangers regulating signaling at various MCS. We propose that in order to unify these disparate observations into a cohesive global understanding of the multiple functions ORP5 and ORP8, future research focusing on the interplay between sterol/phospholipid sensing and transport will be required.

## 2. ORP5/8 as Oxysterol Sensors

Oxysterols are cholesterol metabolites that exert negative feedback on cholesterol synthesis and uptake, as well as influence other biological processes such as apoptosis [6,12]. As OSBP was initially identified as an oxysterol binding protein, early studies on mammalian and *S. cerevisiae* (Osh1-7) ORP family members largely focused on the effects of genetic manipulation of these proteins on cholesterol metabolism [34,35]. Indeed, in yeast, deletion of Osh proteins resulted in a disruption of sterol metabolism [11,36]. The Osh6 and Osh7 ORDs are most closely related to ORP5 and 8 [23,37] (see also Figure 1). Osh6 deletion led to an overproduction and aberrant intracellular distribution of sterols while its overexpression had the opposite effect [11,36,38,39]. It was perhaps these defects in sterol metabolism that lead Suchanek and colleagues to investigate whether mammalian ORPs bound sterols [40]. Both ORP5 and ORP8 bound to the oxysterol photo-25-hydroxycholesterol and to photo-cholesterol in photo-crosslinking experiments, both in vitro and when applied in living cells. Notably, the ORD domains expressed alone was insufficient to mediate sterol binding, and the C-terminal transmembrane domain (residues 266-879) or the full-length protein were required for ORP5 and ORP8, respectively. ORP8 bound strongly and ORP5 moderately to both sterols as compared to other ORPs, although actual binding affinities were not provided. 

For ORP8, numerous studies subsequently pointed to a role in sterol control. Indeed, in vitro binding of its ORD to oxysterols and a role in sterol metabolism was confirmed in macrophages [28]. ORP8 was overexpressed in atherosclerotic plaques and downregulated cholesterol efflux through a transcriptional effect on liver X receptor (LXR) elements within the promoter of the cholesterol transporter ATP binding cassette subfamily A member 1 (ABCA1). In line with a role in oxysterol sensing, in liver cells, ORP8 knockdown and overexpression had opposing effects on cholesterol biosynthesis, and when administered *in vivo*, ORP8 regulated serum lipids levels [41]. The ORP8 ORD was again shown to bind cholesterol *in vitro*, and additionally to be capable of extracting cholesterol from liposomes. The effects on cholesterol correlated with the nuclear localization of sterol regulatory binding protein (SREBP) and were dependent on interaction of ORP8 with and a nuclear pore protein nucleoporin 62 (NUP62). Similarly, ORP8 knockout mice harbored defects consistent with an increased biosynthesis and secretion of high-density lipoprotein (HDL), as well as gender-specific alterations in serum lipid profiles including triglycerides and cholesterol [31]. When bone marrow of ORP8 knockout mice was transplanted into wild-type animals no difference in HDL was observed, indicating the HDL defect maybe liver-specific [42]. Instead an increase was detected in pro-atherogenic very low-density lipoprotein (VLDL) in LDL-receptor knockout model of atherosclerosis. On the other hand, ORP8 knockout transplant recipients had smaller atherosclerotic lesions and more macrophage infiltration, attributed to the fact that ORP8 null macrophages had decreased capacity to form foam cells and produced lower amounts of proinflammatory cytokines. A lipidomics study on a macrophage cell line stably expressing an shRNA against ORP8 provided some clues [43]. Here, under basal conditions cholesterol and cholesterol esters were increased upon ORP8 knockdown. However, when cells were treated with the inflammatory toll-like receptor 4 (TLR4) agonist lipopolysaccharide (LPS), known to increase oxysterol production, there was a reduction in PIP species containing arachidonic acid, the precursor for proinflammatory two-series eicosanoids prostaglandins and leukotrienes. In addition, lipid species containing docosahexaenoic acid (DHA), an omega -3 fatty acid that serves as a precursor for anti-inflammatory resolvins, were higher upon ORP8 knockdown, again predicting that macrophages may have reduced inflammatory potential. It is interesting to note that in this study in addition to sterols, changes in ceramides and especially phospholipids were also detected, providing early hints that ORP8 may have functions beyond oxysterol sensing and cholesterol metabolism. 

Another series of studies that investigated the role of ORP8 as an oxysterol sensor were spurred by the discovery that ORP8 is silenced by miR-143, a micro-RNA associated with obesity, diabetes and cancer [30,44]. Oxysterols had been previously shown to dampen AKT signaling, which normally promotes phosphatidylinositol-3 kinase (PI3K) activity, growth and proliferation, and ORP8 reduction impaired insulin mediated AKT activation in hepatocytes. Oxysterol sensing was presumed to contribute but this was not shown directly. The link between miR-143, ORP8 and AKT signaling was then confirmed in cardiomyocytes [45] and vascular smooth muscle cells [46]. Additionally, in hepatic cells the ability of 25-hydroxycholesterol to reduce mitotic rate was dependent on ORP8 interaction with the spindle protein sperm associated antigen 5 (SPAG5) [47], indicating that ORP8 may display both pro- and anti-proliferative activities. In line with these findings, a subsequent report showed that 25-hydroxycholesterol-induced apoptosis was dependent on ORP8 and particularly on the ability of ORP8 ORD to trigger ER stress [48]. Other more recent studies have confirmed the link between ORP8, ER stress and apoptosis in the context of cancer, suggesting ORP8 to hinder rather than favor cancer progression [49,50], despite its promotion of AKT signaling. However, the exact mechanisms through which ORP8 induces ER stress remain to be described.

In contrast to ORP8, early studies on ORP5 associated its overexpression with cancer progression. In one study, ORP5 was found to be highly expressed in a metastatic pancreatic cancer hamster cell line [29]. Knockdown and overexpression in human and hamster cancer cells revealed ORP5 promotes cell migration, and ORP5 expression correlated with poor prognosis in clinical samples of pancreatic cancer. In a follow-up study, SREBP2 and genes associated with the cholesterol synthesis pathway were found to be upregulated by ORP5 overexpression [51]. SREBP2 upregulation was linked to histone deacetylase 5 (HDAC5) induction, which in turn reduced the expression of the tumor suppressor and phosphatidylinositol-(3,4,5)-trisphosphate (PI(3,4,5)P3) phosphatase and tensin homolog (PTEN). Thus, ORP5 expression lowered PTEN while its knockdown had the reverse effect. In addition, the suppression of cell migration by statins, drugs that block a step of cholesterol synthesis, depended on ORP5 expression. Based on effects on SREBP and levels of the cholesterol precursor hydroxymethylglutaryl coenzyme A (HMG-CoA)-synthase, the authors concluded that ORP5 enhances cholesterol synthesis, although cholesterol levels were not shown directly. Whether oxysterol sensing or cholesterol transport play a role in controlling these functions remains an open question. 

## 3. ORP5/8 in the Control of Vesicular Trafficking

Sterol content has been known to profoundly affect vesicular trafficking and various ORPs including OSBP have been linked to the regulation of secretory and endocytic trafficking [52,53,54,55,56]. Indeed, early yeast studies showed severely impaired endocytosis [36] and defects in polarized exocytosis [57] upon deletion of OSH genes, in addition to altered intracellular sterol distribution. Notably, a yeast-two-hybrid screen identified a novel Osh6 and Osh7 interactor Vps4p, an ATPase involved in multi-vesicular body (MVB) sorting that catalyzes the dissociation and disassembly of endosomal sorting complexes required for transport (ESCRT) complexes from endosomal membranes [38]. Overexpression of residues 366-437 of Osh7, thought initially to be a coiled-coil domain but later shown to be a part of the ORD, and which strongly interacted with Vps4p, led to an MVB sorting defect, a dominant negative effect possibly resulting from Vps4p inhibition. In a parallel study however, the same authors reported that although Osh6 partly colocalized with endosomes, its deletion was not essential for endocytosis, MVB sorting, trafficking to the vacuole (yeast lysosome) or secretion of carboxypeptidase Y to the vacuole. Interestingly, Osh6 bound to PI(4)P, PI(5)P, PI(3,4)P2 and PI(3,5)P2, where both the full-length protein and the ORD domain showed similar binding characteristics, providing another early clue to the role of these ORPs in phospholipid transfer [39]. 

In mammalian cells, a transcriptomic study found surprisingly little effects on lipid-related pathways upon ORP8 knockdown in murine macrophages, and instead revealed major changes related to cytoskeletal and secretory functions [58]. Cells depleted of ORP8 exhibited an abnormal distribution of its previously identified interactor, the nucleoporin NUP62, which enhanced migration and altered microtubule organization [41]. Interestingly, ORP8 was found to compete with the exocyst complex protein Exo70 for NUP62 interaction, suggesting a role of ORP8 in controlling secretory function. Perhaps the most important contribution to our understanding of how sterol control by ORP5 can impact the secretory as well as endocytic pathway came from the pioneering study by the group of Yang, where the critical role of ORP5 in regulating endosomal cholesterol trafficking was discovered [22]. Immunoprecipitation experiments followed by mass spectrometry identified an interaction between Osh6 and Osh7 and the yeast protein Niemann-Pick disease, type C1 (NPC1, Ncr1 in yeast), a key player regulating the exit of LDL cholesterol from late-endosome/lysosomes. This interaction, which was also confirmed in vivo with a split-ubiquitin yeast-two-hybrid assay, was abolished under conditions of sterol depletion. In mammalian cells, co-immunoprecipitation experiments confirmed the interaction between ORP5 and NPC1, but here it did not depend on NPC1 cholesterol binding. Interestingly, the purified ORD of ORP5 (residues 266 to 826) was able to transfer a fluorescent cholesterol analog dehydroergosterol (DHE) between liposomes in vitro, which was partially inhibited by PI(4)P in the donor membrane, providing another early clue to ORP5’s phospholipid-related function. Moreover, knockdown of ORP5, but not of ORP8 resulted in cholesterol accumulation in the limiting membrane of endo-lysosomes, impaired cholesterol transport to the ER and reduced the rate of cholesterol esterification. While ER and lipid droplet morphology were unaltered in ORP5-depleted cells, trans-Golgi network proteins were mislocalized and trapped in cholesterol-filled endo-lysosomal compartments, a phenotype that was not found in ORP8 knockdown cells, but that could be reversed by ORP5 overexpression. Furthermore, knockdown of ORP5 affected trafficking events through early and late endosomes, defects that likely resulted from the accumulation of cholesterol in endosomal compartments. The authors proposed a role for ORP5 in non-vesicular transport of cholesterol from the endo-lysosome limiting membrane to the ER, presumably at ER-endo-lysosome MCS, in response to increased cholesterol and possibly with the assistance of NPC1. This study was perhaps the first groundwork to position ORP5 as a lipid transfer rather than sensor protein, although whether the cholesterol transfer activity in cells was truly direct requires further investigation. 

## 4. ORP5/8 as Phospholipid Transfer Proteins at Membrane Contact Sites

The observation that PI(4)P influenced *in vitro* cholesterol transfer by the ORP5 ORD (residues 266-826) [22], lipidomic data implying a role for ORP8 in regulating phospholipid balance [43], and previous studies on Osh6 and Osh7 showing *in vitro* capturing of various phospholipids including phosphatidic acid (PA), phosphatidylserine (PS), and PI(4)P [39,59], all hinted at a connection between ORP5 and 8 and phospholipids in addition to sterols. However, the first study to show phospholipid capturing by ORP5 was the pioneering report by the group of Gavin, that demonstrated unequivocally the role of Osh6 in transferring PS at ER-plasma membrane (PM) MCS, through a crystal structure of PS within the Osh6 ORD and an elegant series of biochemical and cellular assays [37]. Here, the ORP5 ORD (residues 357-788, a shorter version as compared to Du et al [22]), extracted PS when mixed with liposomes, whereas no other lipids, including cholesterol, were extracted using this method. Together, these observations set the stage for subsequent studies focusing on the role of ORP5 and ORP8 as phospholipid LTPs, first at ER-PM and then at ER-mitochondrial and ER-lipid droplet MCS.

### 4.1. ORP5/8 at Endoplasmic Reticulum-Plasma Membrane Contact Sites

In 2015, two back-to-back papers were published in *Science*, the first focusing on ORP5 and 8 and the second on Osh6, both showing that these ORP homologs transport PS through a PI(4)P counter-flow exchange mechanism at ER-PM MCS [23,60]. In this seminal paper by the De Camilli group, Chung et al. first examined the subcellular localization of ORP5, ORP8L and ORP8S. Confocal microscopy showed GFP-ORP5 almost exclusively at ER-PM junctions in HeLa cells, which differed dramatically from both the endogenous immunostaining and localization pattern of a similar GFP-tagged ORP5 observed previously in the same cell type, which displayed a much more reticular staining and prominent overlap with ORP8L [22]. On the other hand, and similar to this previous report, GFP-ORP8L showed a mainly reticular distribution with only faint ER-PM MCS puncta, whereas GFP-ORP8S had an intermediate localization pattern. ORP5 was suggested to contribute to ORP8 recruitment to ER-PM MCS, as co-immunoprecipitation data showed ORP8L and ORP5 interact, and ORP8L localization at ER-PM MCS was increased upon co-expression with ORP5. These results indicate that in contrast to cholesterol efflux at lysosomes which does not require ORP8 [22], at ER-PM MCS ORP5 and 8 may act in concert. Truncation constructs lacking the PH domains in both ORP5 and ORP8S failed to localize to ER-PM MCS, demonstrating their requirement for PM recruitment. Overexpression of the PI(4)P-synthesizing enzyme phosphatidylinositol-4 kinase III-alpha (PI4KIIIα) increased the cortical pool of ORP5 and ORP8L, while treatment with a PI4KIIIα inhibitor, which reduced PI(4)P but not PI(4,5)P2 levels at the PM led to dispersion throughout the ER. Together these results suggested PI(4)P binding rather than PI(4,5)P2 defines ORP5/8L recruitment. An inducible conditional knockout of PI4KIIIα that showed dramatic loss of PM PI(4)P, some loss of PS but no change in other PIPs, dispersed ORP5 throughout the ER, lending further support. The ORD of ORP8 (residues 370-809) harbored either a PIP or PS as analyzed by mass spectrometry and by quantitative analysis of the lipid fraction extracted from purified ORDs, whereas the ORD of ORP5 could not be purified in sufficient yields. Surprisingly, no other phospholipids, sphingolipids or cholesterol were detected, but this is perhaps consistent with previous observations that the full-length ORP8 protein, rather than the ORD alone, is required, at least for sterol binding [40]. Nevertheless, the ORD of ORP8 alone was able to exchange PI(4)P and PS across liposomes, and overexpression or induced PM recruitment of ORP5, but not of the PI(4)P-binding defective mutant (H514A, H515A), reduced PI(4)P and increased PS at the PM. This process required Sac1 as its knockdown prevented the effect, as did PI4KIIIα inactivation. Furthermore, the authors employed an elegant assay where the PH domain of ORP5 and 8 was replaced with FK506-binding protein 12 (FKBP12, making FKBP12-ΔPH-ORP5 and FKBP12-ΔPH-ORP8) and co-expressed a PM-targeted FKBP rapamycin binding domain (FRB). This allowed rapamycin-induced PM targeting of both ORP5 and ORP8 independently of PM PIP levels, effectively uncoupling the transport activity from recruitment. Addition of rapamycin led to an acute and rapid dissociation of the PI(4)P-binding probe from the PM as well as the recruitment of a PS-binding probe. These results suggest that ORP5 and ORP8 mediate PI(4)P/PS counter-transport between the ER and the PM, helping to control PI(4)P levels while selectively enriching PS at the PM, with dephosphorylation of PI(4)P in the ER by Sac1 powering the lipid exchange in a manner similar to OSBP at ER-Golgi MCS [20]. Moser von Filseck et al. uncovered the same mechanism in yeast, where Osh6 was found to bind PS and PI(4)P specifically and in a mutually exclusive manner, extract PI(4)P and exchange PS for PI(4)P between the ER and PM, all driven by Sac1-mediated dephosphorylation of PI(4)P in the ER [60]. Using mutants that block PS synthesis the authors more clearly demonstrated PS transport and its dependence on a PI(4)P gradient. They showed that only in the presence of Osh6, added lyso PS that was converted to PS was rapidly transported from the ER to the PM, and that this effect could not be observed in cells lacking Sac1. Furthermore, this study provided a crystal structure of Osh6 carrying PI(4)P in its binding pocket, complementing the previously published structure of Osh6 bound to PS [37]. 

The critical role of ORP5 and ORP8 in lipid homeostasis at ER-PM MCS was then further confirmed by the group of Yang, in a study that postulated that PIPs other than PI(4)P might be exchanged for PS transport [26]. Here the authors challenged previous findings, reporting that ER-PM tethering by ORP5 and ORP8 is mediated through the interaction of their PH domain with PI(4,5)P2 rather than PI(4)P. The study first confirmed the difference in subcellular localization between ORP8L and ORP8S observed previously [23]. They were the first to characterize the ORP5B isoform which lacked part of the PH domain, revealing a more reticular distribution for this variant and in part explaining the discrepancy in ORP5 localization mentioned above. After crystallization of the PH domain of ORP8 and structure-based sequence alignments with the PH domains of OSBP, Osh3p and CERT, all of which bind PI(4)P, they found that residues mediating PI(4)P binding were missing in the ORP8 and predicted ORP5 PH domains structures (see also Figure 1). A genetically encoded inducible system termed pseudojanin that makes use of Sac1 and phosphatidylinositol polyphosphate 5-phosphatase type IV (INPP5E) was employed to enable the PM-specific depletion of either PI(4)P or PI(4,5)P2 respectively. Using antibody staining to assess PIP reduction, a dramatic redistribution of GFP-ORP5A from ER-PM MCS to the reticular ER was observed upon depletion of PM PI(4,5)P2 but not PI(4)P, in direct contradiction to previous results [23]. Furthermore, overexpression of phosphatidylinositol 4-phosphate 5-kinase type-1 beta (PIP5K1b), an enzyme that selectively produces PI(4,5)P2 at the PM, increased recruitment of mCherry-ORP8L to ER-PM MCS. These data indicated that PM located PI(4,5)P2, rather than PI(4)P, may be the driving force for ORP5/8 recruitment to ER-PM MCS. Moreover, isothermal titration calorimetry experiments using the purified ORDs of ORP5 and ORP8 and short acyl chain water-soluble PIPs demonstrated binding to all mono-phosphorylated PIPs. These observations are generally consistent with a previous report showing the Osh6 ORD capable of binding PI(4)P, PI(5)P, PI(3,4)P2 and PI(3,5)P2 [39]. However, it is unclear whether short acyl chain PIP derivatives insert into the binding pocket in a similar way to naturally occurring PIPs, particularly since the authors themselves later show, using liposome transfer assays, that acyl chain length can impact the rate of counter-transfer. Nevertheless, the slightly longer ORD of ORP8 (residues 331 to 835) efficiently extracted and transported mono-phosphorylated PIPs, as well as PI(4,5)P2 from model liposomes *in vitro*. Upon overexpression of mCherry-ORP5A, PM PI(4,5)P2 was significantly reduced, while double knockdown of ORP5/8 had the opposite effect, increasing PI(4,5P)2 as well as a decreasing PM PS. Unfortunately, the effect of double knockdown of ORP5/8 on PI(4)P was not assessed. Regardless, these results confirmed the role of ORP5 and 8 in the selective enrichment of PM PS as suggested previously [23,37], and additionally revealed a novel role in the maintenance of PM PI(4,5)P2 homeostasis. Finally, an *in vitro* transport assay using liposomes demonstrated that the ORD of ORP8 efficiently mediated the transport of PS from donor membrane to acceptor membranes in the presence of PI(4,5)P2 and PI(3)P, and to a lesser extent of PI(4)P. This highlighted the importance of a PIP gradient for the transport of PS and suggested also that PI(4,5)P2, and not just PI(4)P could potentially serve as counter exchanger with PS at ER-PM MCS. However, it remains unclear whether an ER-localized 5-phosphatases could act in concert with Sac1, or whether the PI(4,5)P2 concentration gradient would be sufficient to power the exchange mechanism. 

Subsequently, several studies shed additional light on ORP5/8 function at ER-PM MCS. First, in a study focusing on the role of the PS synthase PSS1, the group of Balla reproduced the finding that ORP8L PM localization is reduced when PM PI(4)P is depleted by pharmacological inhibition of PI4KIIIα [61]. In addition, ORP8, and to a much lesser extent ORP5 knockdown reduced the rate of PM PI(4)P depletion upon the same treatment. Then, in a follow-up study, the authors employed the same rapamycin-induced ORP5 and ORP8 PM targeting that was previously used by the De Camilli group [24]. Similarly, addition of rapamycin led to an acute and rapid PI(4)P extraction from the PM, which was even more potent for ORP8, reproducing the results of and lending strong support the model proposed by De Camilli. Furthermore, they found that FK-ORP5 and to a much lesser extent FK-ORP8, were able to extract PS, confirming the ORP5/8 PS transport function. Interestingly, PM PI(4,5)P2 levels were unchanged, suggesting that the effects of ORP5 on PM PI(4,5)P2 observed previously may be indirect [26]. However, it could be argued that the FK replacement somehow inhibited PI(4,5)P2 extraction. Cholesterol levels were unchanged as well, indicating that cholesterol is likely not transported by ORP5 or 8 at the PM, but these results do not necessarily exclude a cholesterol transport function at other membranes. The same inducible Sac1/INPP5E/pseudojanin system as Ghai et al. was then employed to deplete PM PI(4)P, PI(4,5)P2 or both in COS-7 cells, although the PIP levels were assessed using lipid probes rather than antibodies. Here, while PI(4,5)P2 depletion was similarly shown to decrease GFP-ORP5 recruitment, PI(4)P depletion was also effective, in direct contrast to the previous report [26], indicating both lipids are required for ORP5 recruitment. On the other hand, ORP8 was recruited only when PI(4,5)P2 levels were increased through overexpression of PIP5kIb, in agreement with the previous study. Conversely, while not sufficient, PI(4)P was still required for ORP8 recruitment, as pharmacological inhibition of PI4KIIIα reversed recruitment. Additionally, a detailed analysis including localization and liposome binding studies of truncation mutants, as well as nuclear magnetic resonance (NMR) and molecular dynamics simulation for ORP8L, revealed that the upstream polybasic sequence preceding the PH domain of ORP5A and ORP8L influence ORP5/8 PIP binding to both PI(4)P and PI(4,5)P2 and PM recruitment, and that this sequence is also responsible for an inhibitory effect of ORP8L on ORP5 PM recruitment. Indeed, the importance of this polybasic domain for PIP binding of ORP8S was also reported in a structural study published the same month [25]. Finally, through a series of cellular assays employing rapamycin-induced lipid kinase and phosphatase recruitment, PI4KIIIα inhibition, and measurements of PIP clearance rates after these manipulations or during agonist activation, the authors deduced an intriguing model where ORP5 and 8 act in concert to maintain PM PI(4,5)P2 levels within a narrow range by adjusting the availability of PM PI(4)P. Thus, during agonist activation increased PI(4,5)P2 recruits ORP8 to ER-PM junctions, where it joins ORP5 and more efficiently clears PI(4)P thereby limiting PI(4,5)P2 production by removal of its precursor. The abundance and size of ER-PM MCS was also reported to be increased when ORP5 and ORP8 were overexpressed together, but not alone, confirming a synergism between the two homologs [32]. While it is still uncertain whether methodological or biological reasons underlie the discrepancies observed between reports, these critical studies together portrayed ORP5 and 8 function in a completely new light, and it now clear that phospholipid transfer across ER-PM MCS is a major mechanism through which these oxysterol sensors exert their function.

### 4.2. ORP5/8 at Endoplasmic Reticulum-Mitochondria Membrane Contact Sites

As ORP5 and 8 began to be recognized as ER-PM MCS proteins, the question of whether these ORPs could operate at other MCS naturally arose. Indeed, ORP5 and ORP8 also localized to ER-mitochondria MCS and were found to be involved in mitochondrial function [32]. Immunogold electron microscopy revealed that in addition to ER-PM MCS, overexpressed ORP5 and ORP8 were present at ER-mitochondria MCS in HeLa cells. Endogenous ORP5 and ORP8 were also enriched in and could be purified from mitochondria-associated ER membrane fractions. Mass spectrometry analysis of immunoprecipitated ORP5 not only confirmed the interaction with ORP8, but also uncovered their association with a protein of the outer mitochondrial membrane, protein tyrosine phosphatase-interacting protein 51 (PTPIP51), that was previously shown to promote ER-mitochondria MCS through interaction with the ER-localized MCS tether vesicle-associated membrane protein B (VAPB). Co-expression of PTPIP51 and ORP5/8 caused an increase in ORP5- and ORP8- containing ER-mitochondria MCS and consequently a decrease in ORP5 and ORP8 in the cortical ER. Interestingly, the wild-type ORDs of ORP5 and ORP8, but not PS-binding mutants (ORP5-L389D/ORP8-L425D) or the PH domain, were required for their interaction with PTPIP51 and targeting to ER-mitochondria MCS. Depletion of ORP5, and to a lesser extent of ORP8, led to altered mitochondrial morphology as well as a reduced oxygen consumption. At ER-mitochondria MCS newly synthesized PS is shuttled to the mitochondria where it is rapidly converted to phosphatidylethanolamine (PE), a phospholipid known to play a crucial role in maintaining mitochondria tubular morphology and respiratory functions [62]. Thus, conceivably ORP5 and ORP8 mediate transport of PS to mitochondria at ER-mitochondria MCS, promoting mitochondrial respiration, but future studies will be required to verify this intriguing possibility. 

Interestingly, Osh6 and Osh7 were suggested to mediate sterol transfer at ER-mitochondrial MCS [63]. Consistent with previous reports on their sterol regulation [11,36], the two Osh proteins bound cholesterol with significant affinities *in vitro* with Kd values of 820 and 850 nM for Osh6 and Osh7, respectively, and hindered the transfer of sterol to mitochondria upon their deletion. *In vitro* sterol transport occurred in the absence of ATP, and down a concentration gradient from the ER to mitochondria, suggesting that synthesis of PIPs or PI(4)P counter-transport may not be required, although this was not shown directly. Whether direct or indirect sterol transfer by ORP5/8 at mammalian ER-mitochondrial MCS can contribute to morphology and respiration defects would be important to define in future studies. 

Mitochondrial respiration is not only affected by lipid transfer at ER-mitochondrial MCS, but is also linked to Ca^2+^ transfer at these junctions. In an intriguing recent study, Pulli et al. demonstrated that overexpression of ORP5 and to some extent ORP8 led to increased mitochondrial matrix Ca^2+^ during histamine stimulation, while having no effect on cytoplasmic Ca^2+^, suggesting ORP5 and ORP8 regulate Ca^2+^ signaling at ER-mitochondria MCS [64]. However, knockdown of ORP5/8 separately or simultaneously had no effect on stimulated mitochondrial Ca^2+^ uptake. Whether ORP5 and ORP8 may regulate Ca^2+^ signaling through an increase of ER- mitochondria MCS remains to be confirmed, but it is also possible that ORP5/8 control mitochondrial Ca^2+^ indirectly through their role in maintaining PI(4,5)P2 levels at the PM, as described above [24,26]. Indeed, overexpression of ORP5/8, also increased caveolar Ca^2+^ signaling, and ORP5 increased histamine-induced inositol trisphosphate (IP3), a second messenger synthesized by PI(4,5)P2 cleavage that induces ER-Ca^2+^ release [65,66]. Finally, overexpression ORP5/8 increased cell proliferation, adding Ca^2+^ signaling to the list of mechanisms through which these multifunctional proteins can affect cellular function.

### 4.3. ORP5/8 at Endoplasmic Reticulum-Lipid Droplet Membrane Contact Sites

In addition to ER-PM and ER-mitochondria MCS, a recent study by the Yang group reported that ORP5 localizes to ER-lipid droplet (LD) MCS and functions in PI(4)P/PS exchange [67]. Several ORPs were expressed in HeLa cells treated with oleate to induce LD formation and among all ORPs, only ORP5 showed a strong LD association. Fluorescence and electron microscopy confirmed that both ORP5A and ORP5B were enriched at ER-LD MCS. Similar to mitochondrial targeting [32], the localization of ORP5 to LD MCS was dependent on its ORD but not on its PH domain. Knockdown of ORP5 let to an increase in LD size, whereas overexpression of wild-type ORP5A, but not of PS- or PI(4)P-binding mutants produced the reverse effect. These results suggest that the phospholipid transfer activity is required for the targeting to and functional effect on LD size. However, apparently this effect requires oleate loading, as no change in lipid droplet morphology was noted previously in ORP5-depleted cells [22]. Furthermore, in ORP5 knockdown cells more PI(4)P and less PS were detected on LDs, whereas very little PI(4,5)P2 was observed. While this interesting study revealed a novel and unexpected role of ORP5 at yet another MCS, many questions remain. First, it will be interesting to elucidate how exactly PS and PI(4)P participate in the regulation of LD biology, as discussed by Renne and Emmerling [68]. Whether ORP5 operates independently of ORP8 or a small pool of ORP8 is indeed present at ER-LD MCS, and whether the more reticular ORP5B isoform plays a differential role in LD regulation are both questions that merit further study. Notably, ORP8 shows a mainly reticular ER localization at steady state, but still has an important function at ER-PM MCS during signaling. Thus, whether signaling could influence or reveal an ORP8 function at LD could also be of interest. Finally, whether sterol sensing or transfer contributes to the function of ORP5 at LDs should be addressed in future studies.

## 5. Conclusions and Future Directions

Overall, early studies examining the role of ORP5 and 8 and their yeast counterparts Osh6 and Osh7 showed clear indications that these members of the OSBP family participate in the regulation of cholesterol localization and metabolism both at the cellular and organismal levels. However, later investigations focusing on their detailed mechanism of action revealed them to be phospholipid LTPs at various MCS, with surprisingly limited evidence for a direct role in sterol binding or transport. Thus, in order to gain a more comprehensive understanding of how these important lipid regulators orchestrate their activity, a remaining challenge lies in consolidating these disparate observations into a coherent model (see summary in Figure 2).

To this end, it would be paramount to first more solidly establish whether ORP5 or 8 can or cannot directly transport sterols across MCS, and if they can where and under what circumstances. Several studies have provided *in vitro* evidence that both ORP5 and ORP8 are capable of binding oxysterols and cholesterol [22,28,40,41], but only one directly examined binding within a more physiological setting and here transport was not investigated [40]. It is important to reiterate that the ORD of ORP5 and 8 alone was not sufficient to mediate sterol binding in this context [40], which may explain why cholesterol failed to be co-isolated with or extracted by ORP5 and 8 in later studies that employed only ORDs or truncated proteins [23,24,37]. Although compelling evidence for a role for ORP5 in cholesterol exit from endo-lysosomes has been presented [22], the evidence supporting a direct role in transport, namely the *in vitro* DHE transfer activity, was rather weak, displaying very slow transfer rates, and was by no means conclusive. Thus, whether the endo-lysosomal cholesterol accumulation could be an indirect consequence of phospholipid transport still needs to be addressed. The location of action may also be important: while cholesterol transport activity at the PM was not detected by elegant cellular assays [24], it would be interesting to employ these same methods at endo-lysosomes, mitochondria and LDs, where a different lipid environment or cohort of interaction partners could make the difference. Several other parameters could be considered. The first is that the expression of alternative splice forms is often observed, but their differential functions have not been examined in detail. It is possible then, for example that ORP5B plays a greater role at more internal MCS, while ORP8S may be more pertinent when assessing PM activity. Secondly, as the dynamic recruitment of ORP8L to the PM under histamine stimulation exemplifies [24], the detection of sterol binding or transfer may require active signaling that alters lipid composition, triggers post-translational modifications or exposes novel protein interactors. Alternatively,, sterol sensing (i.e. binding) rather than transport could act by altering phospholipid LTP activity. One scenario could be that sterol binding to the outer surface of the ORP5/8 could induce a conformational change that either stimulates or inhibits its phospholipid transfer activity. If sterols bind within the ORD pocket, even if weakly compared to phosphoinositides, they could competitively compete with phospholipids for binding. Another possibility is that sterol binding could stimulate post-translational modifications or alter the interaction with other protein binding partners.

Finally, it may very well be that ORP5/8 do not directly bind or transfer sterols in a physiological milieu. In this case, whether phospholipid transfer could explain earlier observations of the responses to oxysterol or effects on cholesterol transport and metabolism should be re-examined. For example, changes in lysosomal PI(4)P upon ORP5 ablation could impact the recruitment of other ORPs capable of transporting sterols such as OSBP and ORP1L thus indirectly affect sterol transfer from endosomes [69,70]. In addition, Maekawa and Fairn demonstrated that a proper transbilayer distribution of cholesterol requires PS, and thus changes in cholesterol distribution in ORP5-depleted cells could also conceivably arise as an indirect effect of PS alterations [71]. However, changes in total cholesterol in ORP5-depleted cells observed by Du and colleagues [22] might not be simply due to changes in PS, as no changes in total cholesterol was observed in cells with 80% decreased PS upon reduction of PS synthase activity. Finally, it is also tempting to speculate that the tight regulation of PM PI(4,5)P2 by ORP5/8 could contribute to the ORP8 influence on AKT signaling [30,45,46], as this could conceivably also impact the synthesis of PI(3,4,5)P3 by PI3K. Indeed, recently, the depletion of ORP5 or ORP8 reduced PM PS levels resulting in mislocalization of KRas from the PM, an oncogene that requires PM PS for its localization and activity [72]. Together with effects on Ca^2+^ signaling [64], these are both additional mechanisms based on phospholipid transfer through which ORP5/8 may contribute to the control of proliferation, migration and other functions. Regardless of whether they act directly or indirectly, these versatile proteins are important factors in the maintenance of lipid homeostasis and lipid signaling responses, and a deeper knowledge of their intricate biology will surely lead to a better understanding of lipid physiology as well as lipid-related diseases.

## Figures and Tables

**Figure 1 biomolecules-10-00928-f001:**
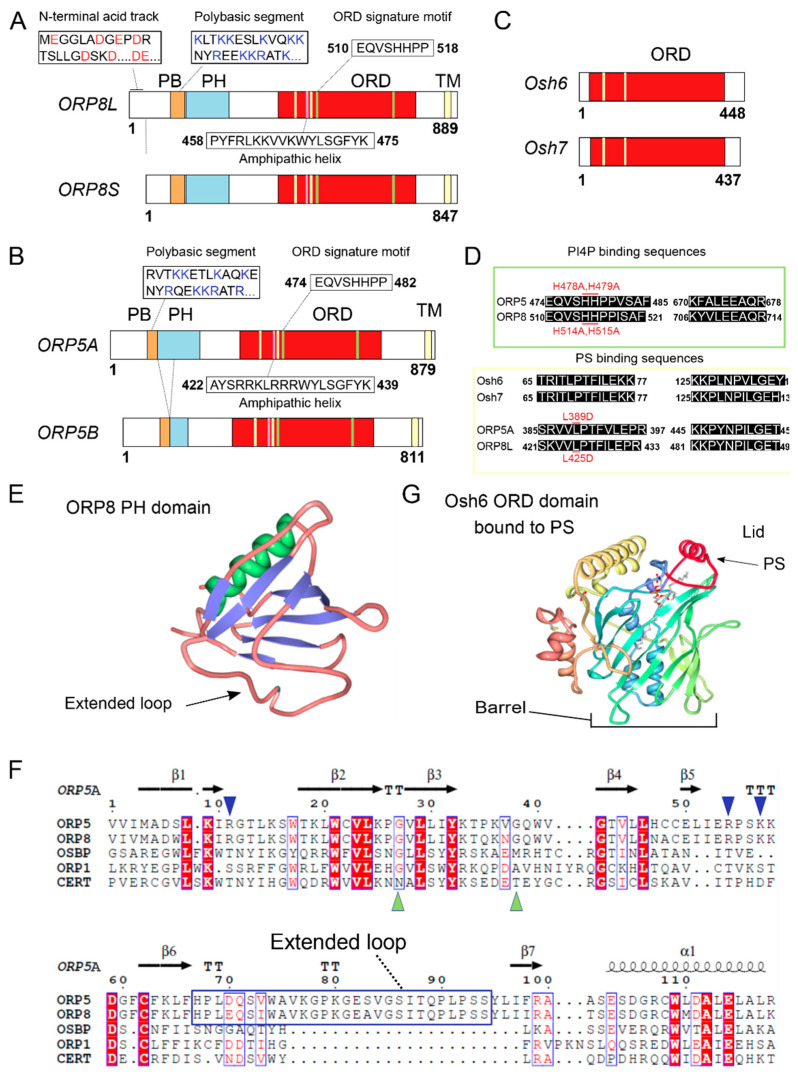
Structure of ORP5 and ORP8 isoforms. (**A**–**C**) Domain structure of (**A**) human ORP8L (canonical form) and ORP8S (missing residues 1-43), (**B**) ORP5A (canonical form) and ORP5B (missing residues 134-201 within the PH domain), and (**C**) *S. cerevisiae* Osh6, Osh7 proteins. Acidic amino acids within the N-terminal domain of ORP8L are highlighted in red. Basic amino acids inside the polybasic segments are highlighted in blue. The position of PI(4)P or PS binding motifs within the ORD domains of ORP5, ORP8, Osh6, Osh7 is indicated by green (PI(4)P) or yellow (PS) strips. The positions of the putative amphipathic helices within the ORD domains of ORP5, ORP8 are indicated by pink strips. **(d)** Residues involved in the recognition of head groups of PI(4)P or PS within the ORD domains of ORP5, ORP8, Osh6, and Osh7. (**E**) Ribbon representation of the crystal structure of the ORP8 PH domain showing alpha helices (green) and β-sheets (blue), as well as the atypical extended loop 3 (arrow) (PDB id: 1V88). (**F**) Multiple sequence alignment of the PH domains from human ORP5A, ORP8, OSBP, ORP1 and CERT highlighting the atypical nature of the ORP5 and ORP8 PH domains. Secondary structure elements of ORP5A are indicated above the alignment. Residues comprising the atypical extended loop present in ORP5A and ORP8 but absent in other PH domains are highlighted by the blue box. Basic amino acids found only in the PH domains of ORP5 and ORP8 proteins but absent in other PH domains are indicated by blue arrows. The amino acids mediating PI(4)P binding which are present in CERT but absent in ORP5 and ORP8 are indicated by green triangles [26]. Alignments were made with ESPript version 3.0 [33]. (**G**) Ribbon representation of the crystal structure of Osh6 ORD bound to PS, the latter shown as a ball-and-stick model (PDB id:4B2Z). The lid covering the barrel is indicated in red. PB: Polybasic segment, PH: Pleckstrin homology domain, ORD: OSBP related domain, TM: transmembrane domain.

**Figure 2 biomolecules-10-00928-f002:**
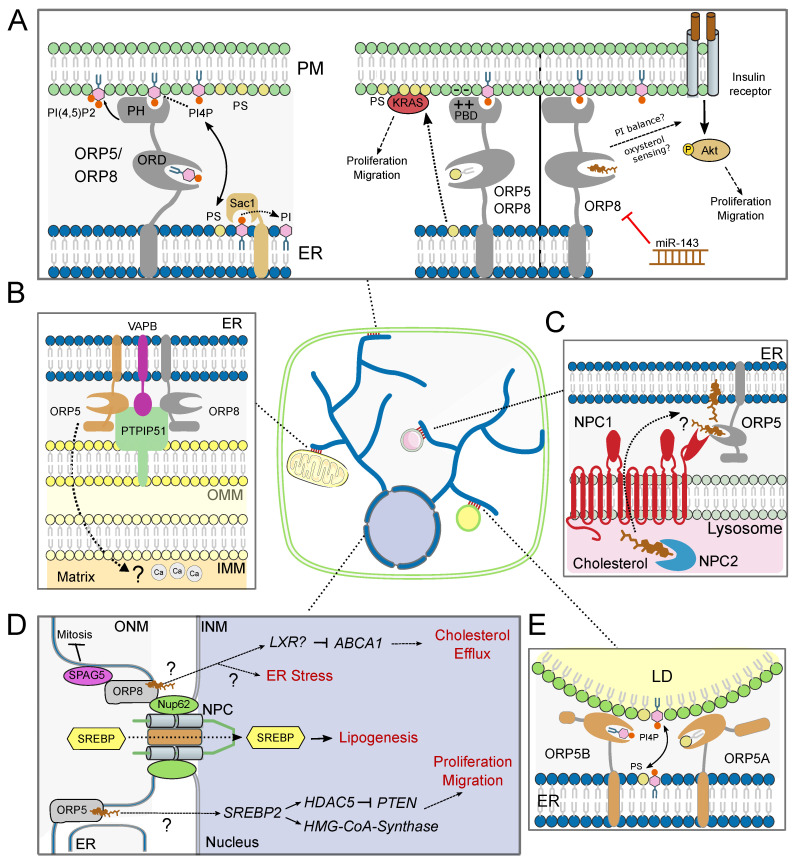
ORP5 and 8 function at various membrane contact sites. (**A**) Endoplasmic reticulum-plasma membrane (ER-PM) membrane contact sites (MCS): ORP5/8 function as phospholipid lipid transfer proteins (LTPs) powered by a counter-flow mechanism, enriching PS at the PM in exchange for PI(4)P which is dephosphorylated in the ER by phosphatase Sac1. Both PI(4)P and PI(4,5)P2 recruit ORP5/8 to the PM through interactions with the PH and PB domains, and PI(4,5)P2 is possibly also transported. ORP5/8-mediated control of PM levels of PS and PIPs may regulate cellular proliferation and migration through the activation of kinases such as KRas and AKT. Sterol sensing may also play a role. ORP8 levels are regulated by miR-143. (**B**) ER-mitochondria MCS: ORP5/8 interact with the mitochondrial MCS tether PTPIP51 (recruited to MCS by interaction with vesicle-associated membrane protein B (VAPB)) and regulate mitochondrial respiration, morphology, and Ca^2+^ signaling, potentially through PS enrichment; (**C**) ER-(endo)-lysosome MCS: ORP5 interacts with cholesterol transporter NPC1 and regulates delivery cholesterol from the limiting membrane to the ER. Whether ORP5 directly transports sterols remains to be confirmed; (**D**) ORP8 interaction with centrosomal protein SPAG5 inhibits mitosis. ORP8 interaction with the NPC protein NUP62 promotes sterol regulatory binding protein (SREBP) translocation to the nucleus, thereby promoting lipogenesis. ORP8 represses cholesterol transporter ATP binding cassette subfamily A member 1 (ABCA1) expression and cholesterol efflux, potentially through liver X receptor (LXR). ORP8 expression also induces ER stress through a poorly defined mechanism. ORP5 promotes SREBP2 expression, inducing hydroxymethylglutaryl coenzyme A (HMG-CoA)-synthase and histone deacetylase 5 (HDAC5) expression, while reducing phosphatase and tensin homolog (PTEN). This axis potentially contributes to increased proliferation and migration. Whether sterol sensing plays a role in these functions remains to be verified; (**E**) ER-lipid droplet (LD) MCS: ORP5A and its shorter isoform ORP5B are recruited to LDs via their ORD, exchange PS and PI(4)P on the LD phospholipid monolayer, and control LD size. PBD: polybasic domain, OMM: outer mitochondrial membrane, IMM: inner mitochondrial membrane, ONM: outer nuclear membrane, IMN: inner nuclear membrane, NPC: nuclear pore complex.

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
